# UBL5/Hub1: An Atypical Ubiquitin-Like Protein with a Typical Role as a Stress-Responsive Regulator

**DOI:** 10.3390/ijms22179384

**Published:** 2021-08-30

**Authors:** Sittinan Chanarat

**Affiliations:** Laboratory of Molecular Cell Biology, Department of Biochemistry and Center for Excellence in Protein and Enzyme Technology, Faculty of Science, Mahidol University, Bangkok 10400, Thailand; sittinan.cha@mahidol.edu

**Keywords:** ubiquitin-like protein, UBL5/Hub1, pre-mRNA splicing, Fanconi anemia pathway, mitochondrial unfolded protein response, stress-responsive regulations

## Abstract

Members of the ubiquitin-like protein family are known for their ability to modify substrates by covalent conjugation. The highly conserved ubiquitin relative UBL5/Hub1, however, is atypical because it lacks a carboxy-terminal di-glycine motif required for conjugation, and the whole E1-E2-E3 enzyme cascade is likely absent. Though the conjugation-mediated role of UBL5/Hub1 is controversial, it undoubtedly functions by interacting non-covalently with its partners. Several interactors of UBL5/Hub1 identified to date have suggested broad stress-responsive functions of the protein, for example, stress-induced control of pre-mRNA splicing, Fanconi anemia pathway of DNA damage repair, and mitochondrial unfolded protein response. While having an atypical mode of function, UBL5/Hub1 is still a stress protein that regulates feedback to various stimuli in a similar manner to other ubiquitin-like proteins. In this review, I discuss recent progress in understanding the functions of UBL5/Hub1 and the fundamental questions which remain to be answered.

## 1. Introduction

Post-translational modification is important for cell growth and stress response. In general, proteins are modified by several types of biochemical reactions, for example, phosphorylation, methylation, acetylation, and ubiquitination. Among them, ubiquitination is unique, because it involves a conjugation of ubiquitin, a 76-amino-acid polypeptide, to other proteins through a reversible isopeptide bond formed between the carboxy (C)-terminal glycine of the ubiquitin and the ε-amino group of lysine or methionine residue of the targets [1,2]. The consequences of ubiquitination depends on whether the ubiquitin is linked to the protein as a monomer or as chains of polyubiquitin [1,2,3]. Particularly, ubiquitin is known for its role in the proteasome-dependent proteolytic pathway, the pathway that degrades the majority of diverse intracellular proteins [1,3]. A vast number of other pathways are also regulated by ubiquitination; such a modification is, thus, crucial for all—molecular, cellular, and organismic—levels of biology [1,2,3].

It has emerged that all eukaryotes express proteins that are related in sequence to ubiquitin and function in an analogous manner. Collectively called ubiquitin-like (UBL) proteins, they play versatile roles in normal and disease-related cellular processes; for example, SUMO regulates transcription, chromatin structure, and DNA repair, RUB1/NEDD8 is implicated in cell cycle control and various cellular pathways, and ATG8 functions in autophagy [1,2,4]. All UBL proteins share a globular β-grasp fold structure composed of a curved β-sheet wrapping around a central α-helix [4,5]. Canonical members of the UBL family contain a di-glycine (di-Gly) motif at their C-terminus for target conjugation and share a capability of being conjugated to target molecules through an ATP-dependent E1-E2-E3 enzymatic cascade, thereby functioning as covalent protein modifiers (Figure 1A,B) [5]. The covalent modification by members of the UBL family alters activities of the substrate by influencing its stability, localization, or behavior of molecular interaction. Depending on the targets, UBL conjugation affects a number of metabolic pathways, for example, DNA damage repair, signal transduction, protein sorting, cellular differentiation, and organ development [4,5].

While the main focus of UBL studies is on canonical members of the family, relatively little is known about functions of UBL proteins which do not covalently modify their target proteins. Here, I review an atypical member of the UBL family, namely UBL5/Hub1, and discuss its molecular functions and future perspectives.

## 2. UBL5/Hub1

Ubiquitin-like protein 5 (UBL5), also known in yeast *Saccharomyces cerevisiae* as homologous to ubiquitin 1 (Hub1), is an evolutionarily conserved 73-amino-acid protein in the UBL family [8,9,10,11]. Similar to ubiquitin and other relatives, UBL5/Hub1 adopts the ubiquitin-like β-grasp fold, the fold that consists of a five-strand antiparallel β-sheet surrounding an α-helix (Figure 1A). Solution and crystal structures reveal that the secondary structure elements of UBL5/Hub1 are β-β-α-β-β-3_10_-β along the protein sequence [9,10,11,12]. According to a Dali search of the Protein Data Bank (PDB), the structure most similar to UBL5/Hub1 among members of the UBL family is ubiquitin, although they share only 22% sequence identity (Figure 1B) [12]. However, unlike ubiquitin, the electrostatic surface of UBL5/Hub1 is highly charged and the protein contains no large hydrophobic patches [9,10,11,12]. From this aspect, UBL5/Hub1 might be a divergent member of UBL family; its structural similarity to ubiquitin may merely reflect a shared folding topology.

Major differences between UBL5/Hub1 and other UBL proteins are at the C-terminus. Typically, most members of the family possess the di-Gly motif, which is processed to achieve the mature form prior to covalent conjugation to protein targets. UBL5/Hub1, however, contains a conserved di-tyrosine (di-Tyr) motif followed by an unconserved amino acid at its C-terminal end (Figure 1A–C). Additionally, UBL5/Hub1 is shorter than the others, as it completely lacks an exposed characteristic C-terminal tail of UBL proteins [8,9,10,11,12].

## 3. To Be (Conjugated), or Not to Be, That Is the Question

As C-terminal tail and di-Gly motif are often required for UBL conjugation, it is controversial whether UBL5/Hub1, which lacks these features, is capable of canonical covalent modification. 

Broadly speaking, UBL proteins fall into two separate categories. Proteins of the first category function as covalent modifiers in a manner analogous to ubiquitin [4,5]. They are processed by specific enzymes and then covalently attached to target molecules via their C-termini. These proteins are often termed “ubiquitin-like modifiers”, or ULMs. On the other hand, proteins of the second category, called “ubiquitin-domain proteins”, or UDPs, bear one or more domains resembling ubiquitin in primary sequence and tertiary structure [4,8,13,14]. Unlike ULMs, UDPs are not conjugated to other molecules and function merely by forming protein complexes or binding to their interactors. Often in the form of modular domain proteins, UDPs are widely encoded in eukaryotic genomes and function in many cellular pathways: Rad23, ubiquilin, and Parkin play roles in proteasome-mediated protein degradation; homocysteine-induced ER protein (Herp) functions in endoplasmic reticulum-associated protein degradation; oligoadenylate synthase-like (OASL), a unique two-UBL-domain protein, is implicated in antiviral activity [13,14].

Originally, a group of scientists proposed in an in vivo study in *Saccharomyces cerevisiae* that UBL5/Hub1 may be a ULM, which is able to be attached to protein targets [15]. In yeast, Hub1 is 73-amino-acid long and contains a di-Tyr motif followed by a leucine residue (Leu73) at its C-terminus. In a western blot analysis of Hub1-immunoprecipitation, Hub1 wild-type does not show any conjugates [15]. However, a mutant variant lacking the Leu73 strongly forms multiple high-molecular-mass species of Hub1-protein adducts, and removal of the last two residues, Tyr72 and Leu73, completely abolishes the complex. Based on these results, the di-Tyr motif is required for modifying protein targets and Hub1 must be post-translationally processed by the removal of Leu73 [15]. Nevertheless, it has not been confirmed whether Hub1 is conjugated to its target via a covalent bond; additionally, Hub1 processing protease as well as Hub1-specific E1-E2-E3 enzyme cascade, if any, are yet to be identified.

Contradictory to the above ULM hypothesis, in vivo and in vitro studies performed by two independent laboratories suggest that UBL5/Hub1 may not be a ULM [8]. Conventionally, UBL proteins are synthesized as inactive precursors and require an initial processing step to expose a C-terminal glycine in the mature modifiers. Specific proteases, for example, deubiquitinating enzymes (DUBs) or UBL-specific proteases (ULPs), are responsible for processing [1,2,3]. Moreover, an ATP-driven E1-E2-E3 enzymatic cascade activates each modifier prior to target conjugation [2]. The norm, however, may not apply to Hub1. Typically, conjugation of ubiquitin and ULMs is sensitive to the thiol-reactive agent *N*-ethylmaleimide (NEM), an inhibitor of DUBs and proteins with active site cysteines, including E1, E2, and E3 enzymes [16]. Experimental evidence shows that not only does NEM fail to inhibit the formation of SDS-resistant high-molecular-weight protein adducts of Hub1, but the drug also even induces elevation of such adducts [8]. Moreover, unlike the mechanisms of other UBL conjugations, Hub1-protein adducts are ATP-independent and insensitive to either EDTA or apyrase [8]. The C-terminal di-Tyr of Hub1 does not seem to be required, as the removal or mutation of the dipeptide does not affect Hub1-adduct formation [8,17]. Notably, an intact C-terminally tagged Hub1 still forms the adducts, suggesting that C-terminal processing of Hub1 is not required for Hub1-protein complex formation [8]. If this is the case, the observed Hub1-containing SDS-resistant adducts are most likely generated by a mechanism dissimilar to the conjugation of ubiquitin and ULMs (Table 1). 

Whether UBL5/Hub1 is a ULM or a UDP, the protein is nevertheless a unique member of the UBL protein family. As a ULM, UBL5/Hub1 is conjugated to targets via di-Tyr instead of typical di-Gly motif; as a UDP, UBL5/Hub1 is the only gene identified to date that encodes only a single UBL domain without any additional module and is most likely the smallest protein of this class [13,14,18].

## 4. Molecular Functions of UBL5/Hub1

While its mode of function as a UBL protein is debatable, several lines of evidence support UDP functions of UBL5/Hub1, which regulates certain cellular pathways via interactions with its binding partners. Major molecular findings on the functions of UBL5/Hub1 in pre-mRNA splicing, the Fanconi anemia pathway, and the mitochondrial unfolded protein response are discussed below.

### 4.1. Pre-mRNA Splicing

Splicing is a fundamental step of RNA processing, involving the removal of intronic sequence in precursor messenger RNA (pre-mRNA) [19,20]. Accurate pre-mRNA splicing is absolutely required for correct gene expression since any error—even at a single-nucleotide resolution—potentially causes aberrant protein products, thereby ultimately resulting in cellular dysfunction and cell death [19,20,21,22]. The reaction of pre-mRNA splicing occurs in the spliceosome, a megadalton ribonucleoprotein complex composed of five small nuclear ribonucleoproteins (snRNPs) and a large number of non-snRNP splicing factors [19,20]. To form a catalytically active spliceosome, snRNPs and non-snRNP-associated factors are sequentially and co-transcriptionally recruited de novo to every single intron of the pre-mRNA. First, the 5′-splice site of the intron is recognized by U1 snRNP. U2 snRNP then binds to the intronic branch-site sequence. Next, the pre-assembled U4/U6.U5 tri-snRNP is recruited and forms a pre-catalytic spliceosome. During each step of spliceosome assembly, several RNA helicases help rearrange conformations of proteins and RNA molecules and activate the spliceosome [23,24]. Suboptimal introns are rejected and discarded by the RNA helicases via kinetic proofreading mechanisms in order to keep splicing fidelity under control [23,24,25,26].

UBL5/Hub1 binds to at least two sites of the spliceosome: the U4/U6.U5 tri-snRNP—via an interaction with splicing factors such as Snu66, Prp38, and Spp381, which are yeast homologs of SART1, PRPF38A, and MFAP1, respectively—and the DEAD-box RNA helicase Prp5 (Figure 2A–C) [10,11,27]. SART1/Snu66 and PRPF38A/Prp38 are proteinaceous components of the U4/U6.U5 tri-snRNP. Depending on the species, the Hub1-interaction domain (HIND) motif resides either in the N-terminus of SART1/Snu66 or in the C-terminal end of PRPF38A/Prp38, or in both proteins of some species, such as *Plasmodium falciparum* [10]. Compared to the interaction between ubiquitin and ubiquitin receptors, UBL5/Hub1 interacts with the HIND motif in a distinct manner. While ubiquitin binds to its receptors mostly via weak hydrophobic interaction, UBL5/Hub1–HIND interaction comprises a strong salt bridge between conserved amino acids, arginine of HIND, and aspartate of UBL5/Hub1, supported by several hydrophobic contacts with high binding affinity [10,11]. As demonstrated in yeast, plant, and mammalian studies, the interaction of UBL5/Hub1 with HIND motif is evolutionarily conserved and is important for the function of this ubiquitin relative, indicating its important role in pre-mRNA splicing [10,11,28]. Intriguingly, transferring the HIND motif from Snu66 to other components of the U4/U6.U5 tri-snRNP fully restores the function of yeast’s Hub1; it is, therefore, believed that the protein serves as a part of the U4/U6.U5 tri-snRNP rather than directly regulating the role of SART1/Snu66 or PRPF38A/Prp38 [10]. Indeed, several structural analyses using single-particle cryo-electron microscopy (cryo-EM) reveal that UBL5/Hub1 resides in different complexes during the process of spliceosome assembly, particularly between pre-catalytic and activated spliceosomal complex formation [29,30,31]. Besides SART1/Snu66 and PRPF38A/Prp38, UBL5/Hub1 also makes a direct contact with several domains of the core splicing factor PRPF8/Prp8 [30,31]. It is important to note that while UBL5/Hub1 binds to SART1/Snu66 during the pre-catalytic spliceosomal complex (B complex), another B-complex-specific protein MFAP1/Spp381 replaces SART1/Snu66 after the complex is activated by the spliceosomal RNA helicase SNRNP200/Brr2 [30]. In cooperation with MFAP1/Spp381, UBL5/Hub1 likely plays a role in stabilizing the position of 5′-exon and the interaction of U5 snRNA within the pre-catalytic and activated spliceosomal complexes [30]. As UBL5/Hub1, SART1/Snu66, and MFAP1/Spp381 are present in foci in close proximity to—but not entirely overlapped with—the Cajal bodies, which is involved in the assembly and recycling of snRNPs [32,33,34,35,36], it is tempting to speculate that they might contribute to the processes. It is also important to note that MFAP1/Spp381 interacts with UBL5/Hub1 via the HIND-interacting surface in a mutually exclusive manner (Figure 2C) [30]. The mechanism and significance of the substitution is, however, still unknown and yet to be investigated. In any case, UBL5/Hub1 is located at a strategic position within the pre-catalytic and activated spliceosome and helps regulate pre-mRNA splicing.

Another spliceosomal interactor of UBL5/Hub1 is Prp5, an evolutionarily conserved RNA helicase in the DEAD-box protein family [27]. Also known as DDX46, it is involved in bridging U1 and U2 snRNPs during early steps of spliceosome assembly [37]. Prp5 also facilitates the binding of U2 snRNP to branchpoint regions of the intron and functions as a splicing fidelity factor through a kinetic proofreading mechanism [25,37,38]. During assembly of the pre-spliceosomal complex, Hub1 binds directly to Prp5 at the invariant aromatic amino acid residue, tryptophan-257 in yeast. The residue is highly conserved and is in a close proximity to the ATP-binding and hydrolysis site of the helicase [27]. Hub1 interacts with the enzyme via the opposite side of the surface that binds to HIND motif of SART1/Snu66 (Figure 2B) [27]. Acting as an ATPase stimulating co-factor, this ubiquitin relative enhances Prp5′s RNA-dependent intrinsic ATPase activity in a dose-dependent manner, thereby facilitating overall splicing efficiency [27]. Cellular levels of Hub1 seem to control Prp5 and affect general splicing. When cells are unstressed, the expression level of Hub1 is much less abundant than other spliceosomal components, suggesting a minimal activation of Prp5 and the spliceosome by Hub1 under normal conditions [27]. Experimental overexpression of Hub1, however, not only activates Prp5 and the spliceosome, but also affects the selection of splice sites, most likely due to faster splicing kinetics; thereby, the stringency of splicing fidelity becomes more relaxed [27]. More importantly—like many other UBL proteins—UBL5/Hub1 is highly expressed under several stress conditions, for example, hypo-osmotic pressure, oxidative stress, and exposure to heavy metals [10,39,40,41]. In budding yeast, the stress-induced overexpression of Hub1 is controlled by stress-responsive regulator Yap1 (for yeast-AP1) at the transcriptional level, promotes intron-specific pre-mRNA splicing, and, as a result, helps facilitate tolerance of such stresses (Figure 3A) [39,42]. 

Hub1-mediated alternative splicing plays physiological roles not only in the stress response but also in the maintenance of integrity of the nuclear envelope in *Saccharomyces cerevisiae*. In this yeast, Hub1 regulates alternative splicing of *SRC1*, also called *HEH1*, which is a gene that encodes an inner nuclear envelope protein [10,43]. Tandem arrangement of overlapping 5′-splice sites in *SRC1* pre-mRNA results in two isoforms of mature mRNA and subsequent expression of two protein variants [10,44]. The long isoform Src1-L is an 834-amino-acid protein and harbors two transmembrane domains and a winged-helix MSC domain (for MAN1-Src1 C-terminal) at its C-terminus. Alternative splicing of the *SRC1* pre-mRNA, however, causes a shift in the open reading frame, resulting in an expression of the shorter isoform Src1-S lacking the second transmembrane and the C-terminal MSC domains [10,43,44]. While both tandem splice sites of *SRC1* pre-mRNA are used in wild-type, cells lacking Hub1 are unable to splice the upstream splice site to produce Src1-S isoform; thereby, only Src1-L isoform is expressed [10]. It has recently been shown that Hub1-mediated splicing generates Src1-S to modulate the recruitment of the endosomal sorting complexes required for transport (ESCRT) machinery to the nuclear envelope, which is critical for the nuclear integrity [43].

UBL5/Hub1 also plays a splicing-mediated physiological role in humans, i.e., sister chromatid maintenance. Loss of UBL5 causes a global reduction in pre-mRNA splicing efficiency and an increase in intron retention [11,45]. It appears that splicing of not all but specific introns are affected; the phenomenon is similar to Hub1′s intron-specific splicing regulation observed in yeasts [10,17,27,46]. Particularly in UBL5-deficient cells, the first intron of Sororin, a cohesion protection factor, is retained, while splicing of other introns of the same gene is unaffected [45,47]. The intron retention results in remarkable loss of Sororin protein and a decrease in the load of the protection factor onto chromatin upon mitotic exit [45]. In sum, UBL5 is required for human cells to prevent premature sister chromatid separation by promoting proper pre-mRNA splicing and expression of the cohesion factor Sororin [45].

In higher eukaryotes, UBL5/Hub1 may add another layer of alternative splicing control through its interacting partners, which belong to a class of SR-protein kinases [9,48]. In most eukaryotes, alternative splicing is significantly regulated by a family of serine/arginine-rich (SR) proteins, which are often phosphorylated by several SR-protein kinases [49,50,51]. Upon hyper-phosphorylation, SR proteins directly interact with the pre-mRNA and stabilize interactions between spliceosomal components during spliceosome assembly [49,52,53]. By contrast, de-phosphorylation of SR proteins is required in the late step of splicing for the nuclear export of mature mRNA [54,55]. One family of SR protein-modifying enzymes is the cell division control protein (CDC)-like kinase (CLK), an evolutionarily conserved kinase family that phosphorylates serine, threonine, and tyrosine residues [56,57]. All four isoforms of CLKs are expressed in most cell types and tissues and affect a vast variety of biological processes via splicing control [58]. Notably, yeast two-hybrid screens reveal that UBL5 interacts with all four members of the CLK family [9,11,48,59]. Though still elusive, UBL5-CLK interaction may provide an additional layer of SR-protein-mediated regulation of alternative splicing. The hypothesis is yet to be proven.

### 4.2. Fanconi Anemia Pathway

While the roles of UBL5/Hub1 in pre-mRNA splicing are widely established, a line of evidence also suggests that this ubiquitin relative is also involved in the Fanconi anemia (FA) pathway, a DNA damage response mechanism that repairs DNA interstrand crosslinks (ICLs) [45,60,61]. ICLs, which are covalent linkages between the opposite strands of double-stranded DNA, can be generated by endogenous metabolites (e.g., aldehydes and nitrous acid) or by exogenous origins (e.g., psoralen or cisplatin used in cancer chemotherapy) [62,63]. The FA pathway comprises 19 Fanconi anemia proteins (FANCA to FANCT) and a number of associated proteins. In response to the damage, FANCD2-I heterodimer and FA core complex are recruited to the DNA lesions once the ICLs are detected [63]. In conjunction with E2 conjugating enzyme FANCT, the FA core complex harboring an E3 ubiquitin ligase activity monoubiquitinates and activates FANCD2-I heterodimer, which subsequently coordinates many downstream DNA repair events [47,62,63,64].

It has been shown in human cells that UBL5 binds to FANCI and promotes the functional integrity of FA pathway. The interaction between these two proteins is constitutive during the S and G2 phases of the cell cycle and is independent of UBL5′s splicing function and of the monoubiquitination state of the FANCD2-I heterodimer [60]. While existing as a heterodimer with FANCD2, FANCI is additionally able to form FANCD2-independent homodimers and/or oligomers [60]. This higher-order formation of FANCI complex requires UBL5, which stabilizes through a direct protein–protein interaction and protects the complex from proteasome-mediated degradation [60]. While UBL5–FANCI interaction is unaffected by the treatment of mitomycin C—an alkylating agent that induces ICLs—the protein levels of UBL5 is slightly increased upon the drug exposure [60]. Because UBL5 is not accumulated at the damaged DNA and the FA-related role of UBL5 does not require its splicing activity, it is believed that UBL5 stabilizes FANCI and facilitates function of the protein before the recruitment to the ICLs, thereby promoting the FA DNA damage response pathway (Figure 3B) [60].

### 4.3. Crosstalk between FA and Pre-mRNA Splicing 

Various lines of evidence show existence of the crosstalk between DNA damage repair and pre-mRNA splicing [64,65,66,67,68,69]. In particular, it has been shown that proteins in the FA pathway regulate the nuclear dynamics of splicing factors [70]. In situ proximity ligation assay verifies that FANCI and FANCD2 are in close proximity with SF3B1, the largest subunit of U2 snRNP and the most frequently mutated splicing gene in cancers [64,70,71,72]. In response to ATR activation, FANCI specifically promotes the mobilization of nucleoplasmic pool of SF3B1 from nuclear speckles [70]. Moreover, both FANCI and FANCD2 promote the timely displacement of SF3B1 and certain splicing factors from the chromatin, thereby organizing the nuclear dynamics of splicing factors [70]. Though there is no direct evidence that shows an implication of SF3B1 in FA pathway, it has been reported that components of U2 snRNP are important for the stability of several DNA damage repair proteins [73]. Since UBL5/Hub1 binds to FANCI and U2 snRNP-associated RNA helicase Prp5, it is tempting to speculate that UBL5/Hub1 may play a role in the crosstalk between these two processes/machineries. 

### 4.4. Mitochondrial Unfolded Protein Response

A large number of mitochondrial proteins are encoded in the nuclear genome and must be imported from the cytoplasm into the organelle with the help of molecular chaperones [74]. Several perturbations during mitochondrial protein synthesis and import result in accumulation of unassembled proteins. Upon such a failure, stress responses are rapidly activated to alleviate the proteostasis defects by modulating the environment of protein folding through regulation of the translation and availability of chaperones. In particular, the mitochondrial unfolded protein response (UPR^mt^) is a stress response system of mitochondria, which selectively activates the transcriptional programs of chaperones and proteases to maintain homeostasis of mitochondrial proteins. 

By RNA interference (RNAi) genetic screen in *Caenorhabditis elegans*, *ubl-5* was identified as a candidate factor that plays a role in UPR^mt^ [75,76,77]. Further verification confirms that inactivation of *ubl-5* is associated with a decrease in expression of *C. elegans* UPR^mt^ genes, *hsp-60* and *hsp-6*, which encode mitochondrion-specific matrix chaperones [75,76]. Expression and activity of *ubl-5* is likely controlled by the insulin/insulin-like growth factor (IGF) signaling pathway because downregulation of *daf-2*, a tyrosine kinase homolog of the mammalian insulin/IGF receptor family, induces upregulation of *ubl-5*. Moreover, the ubiquitin relative becomes overexpressed upon mitochondrial stresses at the transcriptional level under the control of stress-induced transcription factor *elt-2* and *daf-16*, whose activity is negatively controlled by *daf-2*. As a result, *ubl-5* becomes accumulated in the nucleus and executes the late step in gene expression control of the UPR^mt^ (Figure 3C) [75,77]. 

Though the nuclear localization of the ubiquitin relative is consistent with its reported role in pre-mRNA splicing control, it is still unknown whether the protein is implicated in other aspects of gene expression, such as transcription. The precise molecular mechanism of UBL5/Hub1-mediated UPR^mt^ remains to be elucidated.

## 5. Differential Expression of UBL5/Hub1 and Disease Association

Though the loss of *HUB1* gene in budding yeast does not show a growth phenotype under controlled laboratory conditions, mutation of the gene results in growth or development defects and cell lethality in fission yeast, plant, and human [10,11,17,28]. Because UBL5/Hub1 plays regulatory roles in several fundamental processes—pre-mRNA splicing, DNA damage repair, and UPR^mt^—it is tempting to speculate that malfunction of UBL5/Hub1 may potentially be involved in human diseases. To date, even though disease-associated mutation in the protein-coding sequence of UBL5/Hub1 has not been reported, several genetic association studies in mammals suggest that the gene might play a part in energy balance, sleep duration, regulation of fat content, obesity, diabetes, and metabolic disorders [78,79,80,81,82,83,84]. Moreover, gene expression analyses of human, mouse, rat, and bovine tissues indicate that UBL5/Hub1 is found in all major tissues and is highly expressed in main glands of the endocrine systems, including hypothalamus, pituitary, thyroid, and adrenal glands [79,85,86,87,88,89,90,91]. As high levels of Hub1 are able to modify gene expression patterns by modulating intron-specific pre-mRNA splicing in budding yeast [27,39,46], it is reasonable to assume that differential expression of UBL5 in mammalian tissues may control splicing-mediated expression of genes, some of which may encode hormones and endocrine regulatory factors. The precise mechanism and constitutive/stress-induced involvement of UBL5/Hub1 in the aforementioned metabolic processes and diseases are yet to be studied. 

## 6. Future Directions

A growing body of genetic, biochemical, and structural evidence has demonstrated roles of the atypical UBL protein, UBL5/Hub1. A number of questions, however, remain partially or completely unanswered. 

(1) If UBL5/Hub1 is indeed a ULM, what are UBL5/Hub1-specific protease(s), E1, E2, and E3 enzymes? How conserved and stress-responsive is the UBL5/Hub1 conjugation pathway?

(2) What are the precise mechanism and physiological role of UBL5/Hub1-protein adducts?

(3) How does the dynamic protein interaction of UBL5/Hub1 with SART1/Snu66 and MFAP1/Spp381 contribute to spliceosome assembly/activation?

(4) What is a mechanistic explanation for how UBL5/Hub1 stimulates the intrinsic ATPase activity of DDX46/Prp5?

(5) How does co-evolution between UBL5/Hub1 and the spliceosome affect intron variability and genome complexity?

(6) What is the nuclear role of upregulated UBL5/Hub1 in response to UPR^mt^? Is it splicing-mediated regulation or other processes? Is there any crosstalk between different activities of UBL5/Hub1?

(7) What controls the differential levels of constitutively expressed UBL5/Hub1 in a tissue-specific manner? What controls the stress-induced level of UBL5/Hub1 in higher eukaryotes?

(8) At a molecular level, how is UBL5/Hub1—constitutively and/or stress-inducibly—implicated in metabolic processes and disorders as well as other diseases?

In conclusion, UBL5/Hub1 is an atypical member of the UBL family due to its distinct mode of function; nevertheless, the protein is typical in a sense because it is responsive to stresses and plays regulatory roles similar to many members of the family. If there is any interplay between UBL5/Hub1 and its relatives, it would be interesting to see what shall be written in the next chapter of UBL family history.

## Figures and Tables

**Figure 1 ijms-22-09384-f001:**
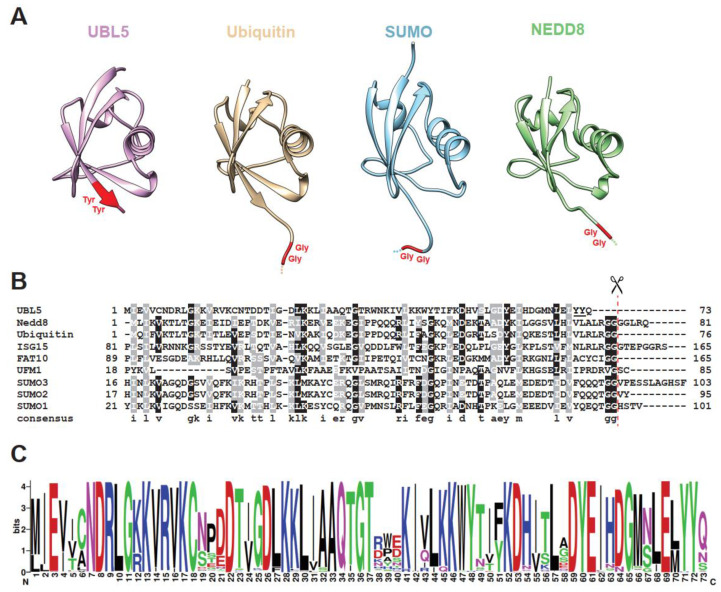
Three-dimensional structures and protein sequences of UBL5/Hub1 and other members of the ubiquitin family. (**A**) Structural ribbon representation of UBL5/Hub1 (PDB: 4PYU), ubiquitin (PDB: 1UBQ), SUMO-3 (PDB: 1U4A), and NEDD8 (PDB: 2N7K). C-terminal di-tyrosine motif of UBL5/Hub1 and di-glycine residues of others are highlighted in red. (**B**) Sequence alignment of human UBL5/Hub1 with ubiquitin and other ubiquitin-like proteins. The scissor symbol shows the processing site after the C-terminal di-glycine motif. The di-tyrosine motif of UBL5/Hub1 is underlined. (**C**) Weblogo [6] representation of the top 250 homologs identified by Blastp search using human UBL5 as query sequence [7]. The *y*-axis represents the bit score. Note that UBL5/Hub1 proteins in certain species carry a few amino acids that are extended from the N-terminus of the core ubiquitin-like domain; therefore, the sequence conservation at the first position is less than the maximum entropy of log_2_ 20 (4.32) bits [6].

**Figure 2 ijms-22-09384-f002:**
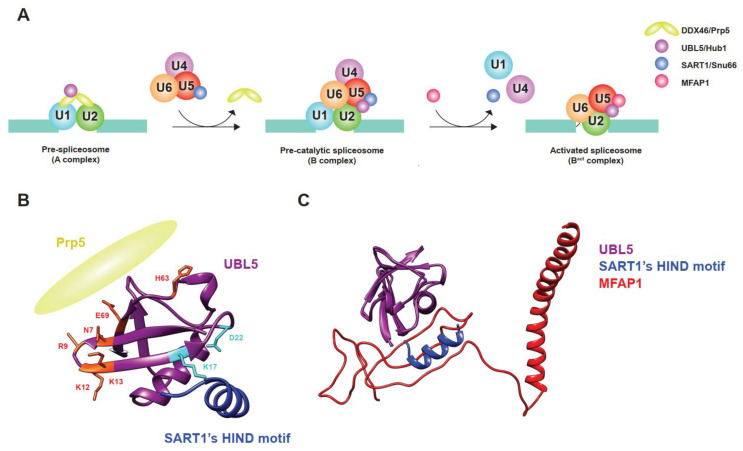
Sequential and differential interaction of UBL5/Hub1 during assembly and activation of the spliceosome. (**A**) Schematic depiction of events in which UBL5/Hub1 functions during assembly and activation of the spliceosome. U1 and U2 snRNPs recognize an intron of pre-mRNA and form a pre-spliceosome (A complex). UBL5/Hub1 binds to the RNA helicase DDX46/Prp5 bridging the two snRNPs. After release of the helicase, the ubiquitin relative binds to SART1/Snu66, a component of U4/U6.U5 tri-snRNP that is recruited during the formation of a pre-catalytic spliceosome (B complex). Subsequent recruitment, release, and rearrangement of proteins and snRNAs lead to formation of an activated spliceosome (B^act^ complex). During the activation, MFAP1/Spp381 replaces SART1/Snu66 and interacts with UBL5/Hub1. (**B**) Ribbon diagram that shows two functional surfaces of UBL5/Hub1 (purple). Amino acid residues that are important for interaction with DDX46/Prp5 (yellow) and HIND motif of SART1/SNU66 (blue) are labeled in red and light blue, respectively. (**C**) MFAP1/Spp381 and SART1/Snu66 share the same surface of UBL5/Hub1 for interaction. Superimposition of human UBL5 structures in the pre-catalytic (PDB: 6AHD) and activated (PDB: 7ABG) spliceosomes show the mutually exclusive interaction of MFAP1 (red) and HIND motif of SART1 (blue) with UBL5 (purple).

**Figure 3 ijms-22-09384-f003:**
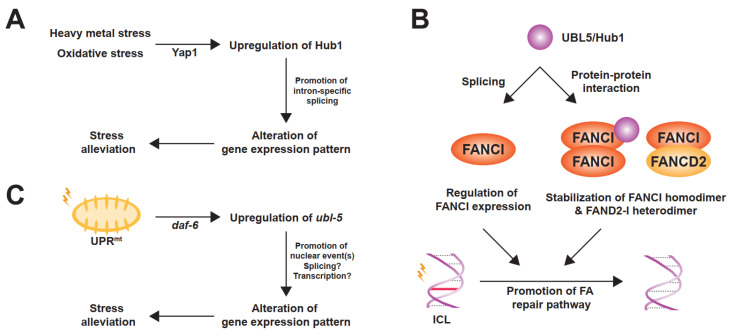
Stress-responsive roles of UBL5/Hub1. (**A**) In *S. cerevisiae*, Hub1 is upregulated at the transcriptional level upon heavy metal and oxidative stresses by the yeast-AP1 (YAP1) transcription factor. Overexpressed Hub1 facilitates intron-specific splicing and, as a result, contributes to stress tolerance. (**B**) As a splicing regulator, UBL5/Hub1 promotes proper splicing of FANCI pre-mRNA. Additionally, this ubiquitin relative interacts with FANCI protein and helps stabilize FANCI homodimer as well as FANCD2-I heterodimer, thereby facilitating the Fanconi anemia (FA) DNA damage repair pathway to resolve interstrand DNA crosslink (ICL). (**C**) In *C. elegans*, mitochondrial unfolded protein response (UPR^mt^) induces upregulation of *ubl-5* which is mediated by the transcription factor *daf-16*. The gene product is then accumulated and functions in the nucleus, though precise molecular function to alleviate the mitochondrial stress is unclear.

**Table 1 ijms-22-09384-t001:** Characteristics of adduct formation/target conjugation with UBL5/Hub1 and other members of ubiquitin-like protein family.

	Requirement for Adduct Formation/Target Conjugation				NEM ^a^ Sensitivity of Adduct Formation/Conjugation
	E1-E2-E3 Enzymes	C-Terminal Glycine	Processing of the C-Terminal Tail	ATP	
**UBL5/Hub1**	No	No	No	No	No
**Ubiquitin and ubiquitin-like modifiers**	Yes	Yes	Yes	Yes	Yes

^a^ NEM: N-ethylmaleimide.

## Data Availability

Data sharing not applicable.

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
