# Peer review of "UBL5/Hub1: An Atypical Ubiquitin-Like Protein with a Typical Role as a Stress-Responsive Regulator"

_ijms, 2021, doi:10.3390/ijms22179384_

Round 1

Reviewer 1 Report

This is a nice review article that discusses the biochemistry and biological role of the ubiquitin-like modifier UBL5/Hub1. Ubl5 is a homologue of ubiquitin with a very similar fold. However, Ubl5 lacks both the flexible C-terminal tail and GlyGly motif normally found on UBLs. Ubl5 is highly charged, which suggests it has its own interaction partners that do not overlap with ubiquitin or UBLs. Its biochemistry does not seem clear as it does not look like it is linked via its C-terminal tail, and the evidence that it is covalently conjugated to proteins is not particularly strong. The summary of the biology seems comprehensive and reads nicely. Ubl5 appears to have an important role in regulating pre-mRNA splicing, and there is some potential cross-talk with the Fanconi Anemia pathway and DNA interstrand cross links. The protein also plays a role in the mitochondrial unfolded protein response. The review is well-written and seems comprehensive. I have no major comments, just a few minor concerns largely with the figures.

Minor comments

In the first paragraph it should be highlighted that ubiquitin can also be transferred to methionine residues. A brief mention of ubiquitin chains should also be included.

Fig 1A: it should be mentioned which SUMO is shown.

Fig 1B: The caption says the TyrTyr is underlined, but it is not.

Fig 1C: what sequences were aligned to form the Weblogo?

Page 3, line 118: delete ‘of level’

Page 4 line 121 seems to contradict with what is stated on page 13 lines 95-106. I suspect this reflects conflicting results from different groups, but this should be better explained.

Table 1: needed? Seems a bit superfluous.

Figure 2B: The author should directly label the proteins / cartoons instead of using a legend. Also, inclusion of the N- and C-terminal residues / amino acid numbers would be appreciated. The turquoise colour does not show up in my copy (it looks more light blue).

Figure 3A and C: these panels should be clarified. The arrow that leads back to the stresses seems out of place.

Reviewer 2 Report

This review article is very nice to understand the Hub1 function and molecular mechanisms.

I think the authors should mention all the UBL  proteins in the introduction to summarize the UBL proteins and their functions.

Furthermore, it would be significant to mention Hub1 mutant phenotype across organisms to understand the physiological significance of Hub1.

And, the relationship between disease and Hub1 is most important to the wide field of readers.

Reviewer 3 Report

In this manuscript, the author has provided a succinct overview of the current literature on the ubiquitin-like protein, UBL5. The review is well-organised, clearly written, and provides direction for future research. Although there are some typographical issues, these should be resolvable during copy-editing.

I believe the article is acceptable as it is.
